# Chemosensory Gene Families in the Oligophagous Pear Pest *Cacopsylla chinensis* (Hemiptera: Psyllidae)

**DOI:** 10.3390/insects10060175

**Published:** 2019-06-17

**Authors:** Ji-Wei Xu, Xiu-Yun Zhu, Qiu-Jie Chao, Yong-Jie Zhang, Yu-Xia Yang, Ran-Ran Wang, Yu Zhang, Meng-Zhen Xie, Ya-Ting Ge, Xin-Lai Wu, Fan Zhang, Ya-Nan Zhang, Lei Ji, Lu Xu

**Affiliations:** 1College of Life Sciences, Huaibei Normal University, Huaibei 235000, China; jwxu32@163.com (J.-W.X.); xyzhuhbnu@163.com (X.-Y.Z.); xuetao_26@163.com (Q.-J.C.); 15556116800@163.com (Y.-J.Z.); yxyang0207@163.com (Y.-X.Y.); 18110329851@163.com (R.-R.W.); zzyy981012@163.com (Y.Z.); a2574138112@163.com (M.-Z.X.); 13030605380@163.com (Y.-T.G.); wxlzy0822@163.com (X.-L.W.); 2Key Laboratory of Animal Resistance Research, College of Life Science, Shandong Normal University, Jinan 250000, China; zhangfan@sdnu.edu.cn; 3Key Lab of Food Quality and Safety of Jiangsu Province-State Key Laboratory Breeding Base, Institute of Plant Protection, Jiangsu Academy of Agricultural Sciences, Nanjing 210014, China

**Keywords:** *Cacopsylla chinensis*, oligophagous pest, chemosensory genes, transcriptome analysis, tissue expression

## Abstract

Chemosensory systems play an important role in insect behavior, and some key associated genes have potential as novel targets for pest control. *Cacopsylla chinensis* is an oligophagous pest and has become one of the main pests of pear trees, but little is known about the molecular-level means by which it locates its hosts. In this study, we assembled the head transcriptome of *C. chinensis* using Illumina sequencing, and 63,052 Unigenes were identified. A total of 36 candidate chemosensory genes were identified, including five different families: 12 odorant binding proteins (OBPs), 11 chemosensory proteins (CSPs), 7 odorant receptors (ORs), 4 ionotropic receptors (IRs), and 2 gustatory receptors (GRs). The number of chemosensory gene families is consistent with that found in other Hemipteran species, indicating that our approach successfully obtained the chemosensory genes of *C*. *chinensis*. The tissue expression of all genes using quantitative real-time PCR (qRT-PCR) found that some genes displayed male head, female head, or nymph-biased specific/expression. Our results enrich the gene inventory of *C. chinensis* and provide valuable resources for the analysis of the functions of some key genes. This will help in developing molecular targets for disrupting feeding behavior in *C. chinensis*.

## 1. Introduction

Insect chemosensory systems interpret and react to various environmental chemical factors, and impact various life cycle processes, including feeding, predator avoidance, and reproductive behavior [1,2]. Recognition of these behaviors frequently involves a series of genes, including those that encode odorant binding proteins (OBPs), chemosensory proteins (CSPs), odorant receptors (ORs), ionotropic receptors (IRs), and gustatory receptors (GRs) [3,4,5]. These proteins participate in extremely complex biochemical reactions in various tissues to ensure accurate delivery of chemical signals [6].

OBPs and CSPs are two families of integral membrane proteins located on the dendrites of olfactory receptor neurons (ORNs), and are abundant in the lymph antennae [7]. These can capture and bind environmental chemical signals [8,9], which are then transferred to ORs or IRs. The first insect OBPs and CSPs were identified in *Antheraea polyphemus* and *Drosophila melanogaster*, respectively [10,11]. OBPs are small and water-soluble extracellular proteins predominantly expressed in the antennae and are used to transmit hydrophobic odor molecules to ORs [12,13]. CSPs are smaller than OBPs and characterized by four conserved cysteines, and exist in almost all olfactory organs and non-olfactory organs, indicating their functional diversity [14].

Chemosensory receptors often contain ORs, IRs, and GRs. ORs are particularly vital in the process of insect and host odor recognition [15,16,17] and have seven transmembrane domain-like G protein-coupled receptors [18]. Two types of ORs are expressed in insect olfactory sensory neurons (OSNs), including conventional ORs and an odorant co-receptor (Orco; formerly called OR83b) that lacks odor sensitivity [19,20]. ORs can combine with most environmental odorants encoded by a trans-fiber pattern to activate OSNs [21,22]. A specialized OR of *D. melanogaster* is activated by specific odors that trigger an innate response [23]. IRs used to sense environmental chemicals belong to an evolutionarily older class of receptors, including a newly-discovered protein family operating as conserved ligand-gated ion channel receptors referred to as ionotropic glutamate receptors (iGluRs) [24,25,26], as well as divergent IRs [27]. There are likewise various antennal IRs [28,29], indicating that IR families play different roles in the process of insects’ recognition of external information. GRs can recognize various materials such as salts, CO_2_, sugars, and organic compounds [30], and are involved in the feeding activities of insects [31,32,33]. In recent years, various chemosensory genes have been discovered in a number of insect species, including *Adelphocoris suturalis* [34], the sweet potato weevil *Cylas formicarius* [35], *Bemisia tabaci* [36], and *D. melanogaster* [37], using RNA sequencing (RNA-seq) technology as an effective method to identify such genes.

*Cacopsylla chinensis* (Yang & Li 1981), belonging to the Psyllidae family among the Homoptera, grows in temperate and subtropical regions of the world [38], including China and Japan, and has become one of the main pests of pear trees in China [39]. *C. chinensis* is an oligophagous pest that can damage buds and fresh leaves and infect healthy pear trees with decaying disease [40]. The nymphs of *C. chinensis* also secrete a mucus that supports various bacterial growth, affecting pear quality [41]. No molecular-mediated chemosensory behaviors in *C. chinensis* have yet been defined. Targeting pests using chemosensory disruptors can interfere with the insects’ ability to find hosts, feed, and reproduce, and a thorough definition of chemosensation in *C. chinensis* could suggest novel strategies for controlling this pest. In this study, we aim at identifying such potential target genes related to insect chemosensation. We first conducted head transcriptome analysis of *C. chinensis* and then further analyzed the phylogenetic trees and examined tissue expression of the chemosensory genes. It provided the basis for proteomics analysis [42] and searching the potential control strategies [43].

## 2. Material and Methods

### 2.1. Tissue Sample Collection

The morphological picture of *C. chinensis* is shown in Figure 1. Three hundred *C. chinensis* (100 males, 100 females, and 100 nymphs) were collected from Dangshan (34°27’42.62” N, 116°31’40.72” E), Suzhou, China. The 300 heads were dissected out under a stereomicroscope with a sterile scalpel and immediately mixed into liquid nitrogen and sent to Genepioneer Biotech Corporation (Nanjing, China) for RNA-seq. For quantitative real-time PCR (qRT-PCR) analysis, we separately collected 60 males, 60 females, and 60 nymphs divided into heads only or bodies only. They were separately named as MH, male head; MB, male body; FH, female head; FB, female body; NH, nymph head and NB, nymph body and placed in nuclease-free centrifuge tubes as one replicate. Each replicate was repeated three times. All samples were preserved in liquid nitrogen until RNA extraction.

### 2.2. Total RNA Extraction and cDNA Synthesis

We used the MiniBEST Universal RNA Extraction Kit (TaKaRa, Dalian, China) to extract total RNA following the manufacturer’s instructions. RNA quality and concentration were analyzed using an ultramicro-spectrophotometer (MD2000D, Biofuture, UK). The RNA was stored at −80 °C prior to use. Single-stranded cDNA templates were synthesized using 1 μg total RNA from various tissue samples, using primers from the PrimeScript^TM^ RT Master Mix (TaKaRa, Dalian, China). 

### 2.3. Library Preparation for Transcriptome Sequencing

For sequencing, total RNA was extracted using the TRIzol^®^ reagent (Tiangen Biotech, Beijing, China) [44,45] and checked for quality. The mRNA was enriched using magnetic beads, then fragmented randomly by adding fragmentation buffer. First-strand cDNA was synthesized using mRNA templates and a random hexamer and the second cDNA chain was synthesized by adding buffers, dNTPs, RNAse H, and DNA polymerase I. The cDNA was then purified using AMPure XP beads (Beckman Coulter, Beverly, MA, USA). Purified double-strand cDNA was used for end repair, adding poly-A tails and adapters [46]. Finally, cDNA libraries were obtained by PCR. Library concentration and insert size were detected using a Qubit^®^ 2.0 Fluorometer (Life Technologies, Grand Island, NY, USA) and an Agilent Bioanalyzer 2100 system. Quantitative qRT-PCR was used to accurately quantify the effective concentration of the library to ensure its quality. Finally, the cDNA library was sequenced using an Illumina Hiseq 4000 platform with the paired-end (PE) and the sequencing read length was PE150bp.

### 2.4. De Novo Assembly and Unigene Annotation 

After using sequencing by synthesis (SBS) technology, raw reads were cleaned by removing adaptor reads, ambiguous reads (‘N’ > 10%), and low-quality reads (that is, where more than 50% of bases in a read had a quality value Q ≤ 5) using Perl script. Clean reads were then de novo assembled using the Trinity program v2.4.0 (http://trinityrnaseq.sourceforge.net/) with default parameters [47], and the minimal contig length is 301 bp. The reads of *C. chinensis* have been deposited (accession number: SRA9127897). BLAST (http://blast.ncbi.nlm.nih.gov/Blast.cgi) was used to compare Unigene sequences with databases including Nr (non-redundant database) [48], Swiss-Prot [49], KEGG (Kyoto Encyclopedia of Genes and Genomes) [50], KOG (EuKaryotic Orthologous Groups) [51], and COG (Clusters of Orthologous Groups) [52] to obtain amino acid sequences of Unigene genes. Then, HMMER [53] software was used to compare with the Pfam [54] database (including chemosensory proteins families in different insects) to obtain the annotation information of Unigene. 

### 2.5. Phylogenetic Analysis

Phylogenetic trees were constructed using amino acid sequences of candidate genes including CchiOBPs, CchiCSPs, CchiORs, CchiIRs, and CchiGRs of *C. chinensis* (Appendix A), and the signal peptide sequences of OBPs and CSPs were predicted by SignalP 4.1 server [55] with default parameters and then removed. The sequences were aligned by using ClustalX 2.0 (University College Dublin, Dublin, Ireland), and phylogenetic trees were constructed using PhyML [56] based on an LG substitution model [57] with nearest-neighbor interchange (NNI). Branch support was estimated using a Bayesian-like transformation of the aLRT (aBayes) method. Dendrograms were edited using FigTree software (http://tree.bio.ed.ac.uk/software/figtree/).

### 2.6. Quantitative Real-Time PCR

We used qRT-PCR to determine the relative expression levels of chemosensory genes in different tissues (MH, MB, FH, FB, NH, and NB). The experiment was performed in a LightCycler^®^96 (Roche Diagnostics Gmbh, Basel, Switzerland) according to the minimum information for publication of qRT-PCR experiments [58], using a total mixture of 10 μL with 5 μL 2X SYBR Green PCR Master Mix (YIFEIXUE BIO TECH, Nanjing, China), 0.2 μL paired primers designed using Beacon Designer 7.9 (PREMIER Biosoft International, CA, USA) (Appendix A) to check for the absence of primer-dimer peaks, 3.6 μL nuclease-free water, and 1 μL cDNA. The amplification step was executed using a degeneration step at 95 °C for 10 min, followed by 40 cycles of 95 °C for 15 s and 60 °C for 60 s. The melting curve detected a single primer-specific peak, using 93 °C for 30 s and 60 °C for 45 s. A negative control was created using distilled water instead of cDNA template for each test run.

*CchiGAPDH* (glyceraldehyde-3-phosphate dehydrogenase) and *CchiEF* (elongation factor 1-alpha) (Genbank numbers CchiGAPDH: MK940861, CchiEF: MK940862) were used as internal reference genes to calculate the expression of different genes in different tissues. This was carried out using the Microsoft Excel-based software Visual Basic using the Q-Gene method [59]. For six samples, each biological sample was repeated three times in a LightCycler^®^ 96 multiwell plate.

### 2.7. Statistical Analysis

Relative expressions of chemosensory genes (mean ± standard error) were compared using one-way ANOVA in SPSS 21.0 software (SPSS Inc., Chicago, IL, USA), according to the least significance difference (LSD). Differences were regarded as significant at *p* < 0.05. The GraphPad^TM^ Prism 7.0 software (GraphPad Software Inc., San Diego, CA, USA) was used to perform the figures [60]. 

## 3. Results 

### 3.1. Transcriptome Sequencing and Assembly 

We generated a total of 32,879,148 clean reads from a cDNA library using transcriptome sequencing (Table 1). The percentage of reads with Q20 and Q30 quality scores was 99% and 91.25%, respectively. De novo assembly yielded 63,052 Unigenes of high assembly integrity with a mean length of 64,826 bp and a maximum length of 20,509 bp.

### 3.2. Homology Analysis

Blastx homology searches of all 63,052 Unigenes showed that 22,963 (36.41%) had homologous genes in the non-redundant (Nr) protein database with a cut-off E-value of 10^−5^. The best match percentage (58%) was with *Diaphorina citri* sequences, followed by sequences from *Acyrthosiphon pisum* (3%), *Zootermopsis nevadensis* (4%), *Tribolium castaneum* (3%), and others (32%) (Figure 2).

### 3.3. Non-Receptor Chemosensory Gene Families

#### Odorant Binding Proteins (OBPs)

A total of twelve putative OBP genes were identified and named as *CchiOBP1-12*. The phylogenetic trees of the *CchiOBPs* were constructed using three Hemipteran species, including *D. citri, Aphis gossypii*, *Sogatella furcifera*, and Diptera *Drosophila melanogaster* (Figure 3). The results show that *CchiOBPs* had several putative one-to-one orthologous relationships with these species, and only CchiOBP1 belongs to the Plus-C subgroup; the other CchiOBPs were clustered in the Classic subgroup. Among the *CchiOBP* genes, six *CchiOBPs* (*CchiOBP1*, *CchiOBP4*, *CchiOBP5*, *CchiOBP9*, *CchiOBP10*, and *CchiOBP12*) had full open reading frames (ORFs) to encode 104 to 238 amino acids and had signal peptides at the N-terminus (Table 2). The expression profiles of all *CchiOBPs* showed that all *CchiOBPs* were highly expressed in the heads (Figure 4). Among the *CchiOBPs*, five *CchiOBPs* (*CchiOBP1*, *3-6*) were more highly expressed in female heads (FH) than in male heads (MH) (*p* < 0.05). *CchiOBP12* was highly expressed in the heads of both male and female adults. Three *CchiOBPs* (*CchiOBP7*, *CchiOBP10*, and *CchiOBP11*) were highly expressed in male heads, with *CchiOBP7* exhibiting the highest expression level. Additionally, two *CchiOBPs* (*CchiOBP2* and *9*) exhibited significantly higher expression in nymph heads (NH) than other tissues, and *CchiOBP8* showed specific expression in nymphs.

### 3.4. Chemosensory Proteins (CSPs)

In all, we identified eleven transcripts encoding putative CSPs in *C. chinensis*. Seven were full-length genes with predicted signal peptide sequences (Table 2) and had four conserved cysteines in corresponding positions. All CchiCSPs shared 37–89% amino acid identity with other Hemipteran insects. The phylogenetic tree results indicate that all CchiCSPs were distributed on various branches, with one *D. citri* ortholog distribution on each branch (Figure 5). The qRT-PCR results of all *CchiCSPs* showed that *CchiCSP1*, *CchiCSP3*, and *CchiCSP8* were highly or specifically expressed in nymph heads. *CchiCSP2* had adult head-biased expression, but *CchiCSP4* displayed adult body-specific expression. *CchiCSP9* and *CchiCSP10* were significantly highly expressed in male and female bodies, respectively (Figure 6).

### 3.5. Chemosensory Receptor Gene Families

#### Odorant receptors (ORs), ionotropic receptors (IRs), and gustatory receptors (GRs)

We identified 13 distinct Unigenes putatively as seven ORs, four IRs, and two GRs (Table 2). Phylogenetic trees for CchiORs (Figure 7), CchiIRs (Figure 8), and CchiGRs (Figure 9) were constructed using amino acid sequences from *C. chinensis* and other insects. We found that the CchiORs were highly homologous to *D. citri*, and CchiOR2 belonged to the Orco family that is widely expressed and highly conserved in insects. In the phylogenetic tree of CchiIRs and CchiGRs, we found that all *CchiIRs* belong to the antennal IRs, but CchiGRs were not clustered in the CO_2_ or sugar receptor families. Tissue expression analysis showed that almost all were more highly expressed in adults than in nymphs, although *CchiOR6* was specifically expressed in the male bodies of nymphs (Figure 10). *CchiOR1* and *CchiOR2* were more highly expressed in adult heads compared to other tissues, and *CchiOR3* and *CchiOR4* were more expressed in male heads and bodies. Meanwhile, *CchiOR5* was more abundant in nymph and female heads. *CchiIR1* and *CchiIR3* were more highly expressed in nymph bodies and female bodies, respectively, while *CchiIR2* showed adult head-biased expression. However, both *CchiGR1* and *CchiGR2* were highly expressed in nymph heads (Figure 10).

## 4. Discussion

As an important olfactory and taste organ, insect heads are important in communication and feeding behavior [61]. Therefore, the head has always been important in insect research, and a certain number of chemosensory genes have been found in the heads of different insects, such as *S. furcifera* [61], *Tomicus yunnanensis* [62], and *Mythimna separate* [63]. This suggests that chemosensory genes in insect heads should be associated with the above-mentioned chemosensory behaviors. In this study, we first sequenced and analyzed the head transcriptome of *C. chinensis*, and identified 36 chemosensory genes (12 OBPs, 11 CSPs, 7 ORs, 4 IRs, and 2 GRs), indicating *C. chinensis* can also regulate the corresponding behavioral response with the chemosensory genes of the head, just like the insects mentioned above. However, the number of chemosensory genes of *C. chinensis* is less than that of other Hemipteran insects, including 9 OBPs, 9 CSPs, 45 ORs, and 14 IRs of *A. gossypii* [64,65]; 9 OBPs, 12 CSPs, 46 ORs, 35 IRs, and 20 GRs of *D. citri* [66]; 12 OBPs, 9 CSPs, 63 ORs, and 14 IRs of *S. furcifera* [67,68]; and 11 OBPs, 17 CSPs, 50 ORs, and 10 GRs of *Nilaparvata lugens* [69,70]. This may reflect the fact that *C. chinensis* is an oligophagous insect and does not need as many chemosensory genes to recognize host volatiles.

The tissue expression results showed that all *CchiOBP*s were highly expressed in heads, indicating their involvement in the olfactory and gustatory processes of *C. chinensis*. Among the *CchiOBPs*, five *CchiOBPs* (*CchiOBP1*, *3–6*) and two *CchiOBPs* (*CchiOBP7, 11*) were significantly highly expressed in the heads of females and males, respectively. Previous studies have revealed the different functions of OBPs in the chemosensory organs of insect heads. Female mosquitoes of *Culex quinquefasciatus* use CquiOBP1 to detect oviposition attractants [71,72], and male moths of *Spodoptera litura* recognize the female sex pheromones by pheromone binding proteins (SlitPBPs) [73], suggesting that insect OBPs participate in sex-related chemosensory behavior, and the seven CchiOBPs may have similar functions, i.e., regulating oviposition or mating behavior by detecting different pear volatiles or intraspecific pheromones. Additionally, two *CchiOBPs* (*CchiOBP2* and *9*) were significantly highly expressed in nymph heads. The highly expressed *NlugOBP3* in nymphs of *N. lugens* is involved in nymph olfaction on rice seedlings as determined by integrating RNAi and ligand binding assays [74]. Thus, the two CchiOBPs may help *C. chinensis* to complete the feeding process on pear trees by detecting the pear volatiles. Interestingly, we found that *CchiOBP8* was specifically expressed in both nymph heads and bodies, suggesting that it may take part in the feeding and/or other physiological processes of nymphs. The OBP phylogenetic tree showed that most CchiOBPs were adjacent in the phylogenetic tree to DcitOBPs of *D. citri*, indicating that the two species have high homology and some OBPs might have similar functions. The CchiOBP1 was clustered into the plus-C OBP subfamily, and others belonged to the classic subfamily, which is similar to *D. citri* and *S. furcifera* [66,68], suggesting that the mechanism of functional differentiation of OBPs in Hemiptera insects may be the same. 

Among all CchiCSPs, six *CchiCSP*s displayed head-biased expression, suggesting that these genes may have similar functions to the high expression of *CchiOBPs* in the head. The functions of CSPs expressed in important chemosensory organs of the head have been reported in different insects, such as SfurCSP5 of *S. furcifera* has high affinities for three rice volatiles (2-tridecanone, 2-pentadecanone, and β-ionone) [75], and three host volatiles [(Z)-3-hexen-1-ol, (E)-2-hexen-1-al, and valeraldehyde] have a high binding affinity with AlinCSPs (1, 2, and 3) of *A. lineolatus* [76], and SinfCSP19 of *Sesamia inferens* is able to bind sex pheromones and host plant volatiles [77]. These results suggest that this type of insect CSP may have similar functions as OBP in recognizing host volatiles and sex pheromones. In addition, many insect CSPs are broadly expressed in different tissues (leg, gut, and pheromone gland) [78,79,80,81,82,83] and have non-chemosensory functions; for example, SexiCSP3 is involved in survival and reproduction in *S. exigua* [84], AmelCSP5 is crucial in the formation of embryonic integument in *Apis mellifera* [85], and some midgut-expressed SlitCSPs of *S. litura* may have functional roles in the specialization and adaption to different ecosystems [82]. In our study, five of 11 *CchiCSPs* in *C. chinensis* were widely expressed in all investigated tissues, suggesting that these *CchiCSPs* may be involved in governing host plant choice and other physiological behaviors. According to our phylogenetic tree of CchiCSP genes, CSPs from *C. chinensis* and *D. citri* were grouped in the same clade, indicating that some CSP genes may have similar functions between the two species.

Chemosensory receptors stimulate corresponding physiological behavior of insects [86,87,88], and identification of chemosensory receptors is a key step in interpreting mechanisms of insect chemosensation. In this study, the OR phylogenetic trees showed that CchiOR2 belongs to the Orco family, which is necessary for locating other conventional ORs on dendritic membranes and for odor detection [89], and which indicates that *C. chinensis* shares chemosensory mechanisms with other insects [90,91]. Seven of 13 chemosensory receptor genes were highly expressed in the heads. Among them, two conventional *CchiORs* (*CchiOR1* and *4*) were more abundant in male heads than in other tissues, indicating that they may participate in recognizing odorants such as pheromones related to male physiological activities. This is similar to the pheromone receptor (PR) of moths, which specializes in the recognition of female sex pheromones [7,92,93]. Previous studies revealed a female-biased AlucOR46 of *Apolygus lucorum* tunes to some plant volatiles [(S)-(−)-limonene, (R)-(+)-limonene, (E)-2-hexenal, (E)-3-hexenol, 1-heptanol, and (1R)-(−)- myrtenol)] [94], three female-biased ORs of *B. mori* (OR19, 45 and 47) account for some of the female-specific odorant responses, such as oviposition cues (plant volatiles: linalool, benzoic acid, 2-phenylethanol and benzaldehyde) and/or detection of an as yet unidentified male-produced sex pheromone [95,96], and larvae-expressed BmorOR56 has high selective tuning to cis-jasmone, a key mulberry leaf volatile [97]. Thus, *CchiOR5*, which was highly expressed in both female and nymph heads, may have similar roles in *C. chinensis*. Mang found that larvae-expressed BmorGR9 is involved in the promotion of feeding behaviors [98], indicating that *CchiGR1* and *2*, which were highly expressed in nymph heads, may also participate in the feeding progress of *C. chinensis*. BmorGR6, which exists in different tissues (midgut, central nervous system, and oral sensory organs), is not only a taste receptor but also a chemical sensor for the regulation of gut movement, physiological conditions, and feeding behavior of *B. mori* larvae [99]. The highly expressed foregut-specific HarmGR9 of *H. armigera* may contribute to the regulation of larval feeding behavior [100], and some SlitIRs of *S. litura* were detected in proboscises, legs, abdomens, and reproductive tissues and may have diverse functional roles in olfaction, taste, and reproduction [101]. We also found widely-expressed genes such as *CchiOR7* and *CchiIR4*, in contrast to those highly or specifically expressed in female and/or nymph bodies (*CchiIR3*, *CchiIR1*, and *CchiOR6*), indicating these five chemosensory receptors may play a vital role in regulating feeding and other physiological behaviors. The exact function of these genes could be further analyzed by integrating in vivo and in vivo methods [92,93,102].

## 5. Conclusions

In conclusion, we identified an extensive set of chemosensory genes that may be related to the chemosensory and gustatory behaviors of *C. chinensis* by sequencing and analyzing head transcriptomic data. As a first step towards understanding their functions, we comprehensively compared phylogeny and tissue expression and demonstrated male head, female head, and nymph-biased specific/expression genes. Our results provide a valuable resource for analyzing the function of key genes in developing effective biological control agents, as well as helping to describe the chemosensory system in *C. chinensis* and other oligophagous pests.

## Figures and Tables

**Figure 1 insects-10-00175-f001:**
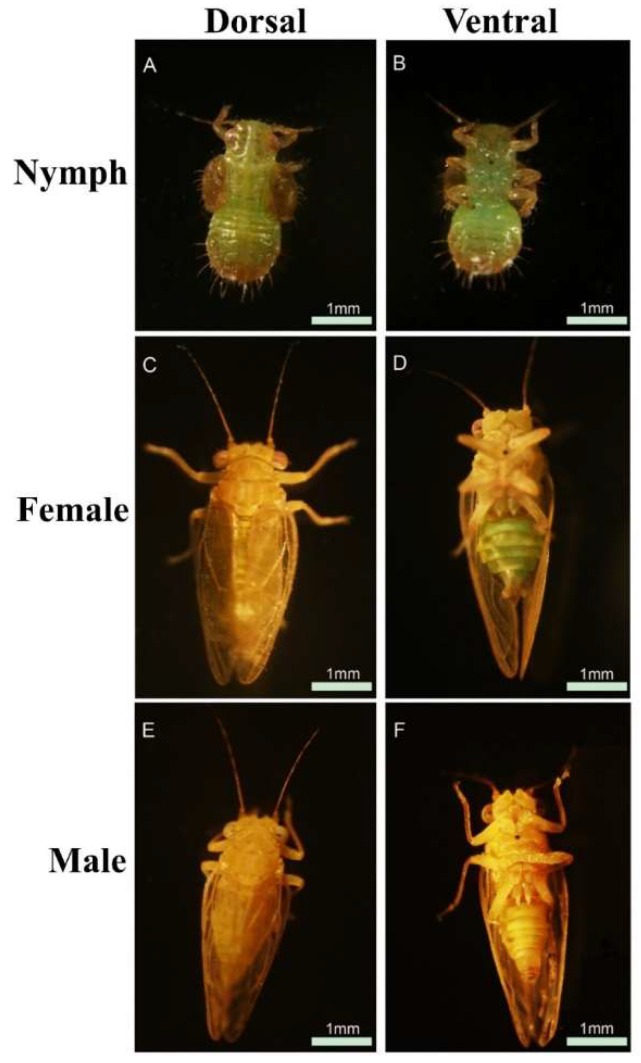
Photographs of *Cacopsylla chinensis*.

**Figure 2 insects-10-00175-f002:**
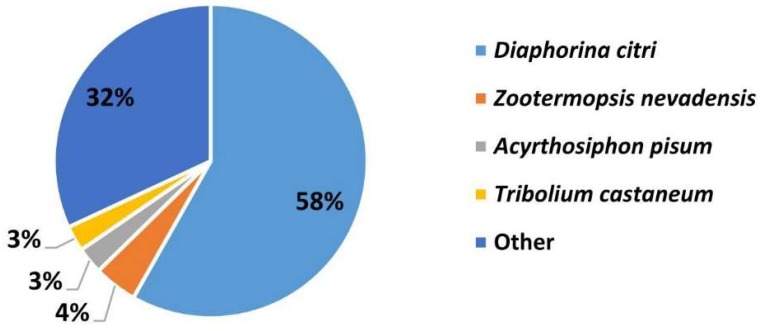
Percentage of homologous hits of the *C. chinensis* transcripts to other species. Blastx searched the *C. chinensis* transcripts against the non-redundancy protein database with a cut-off E-value of 10^−5^. Species that have more than 3% matching hits to the *C. chinensis* transcripts are shown.

**Figure 3 insects-10-00175-f003:**
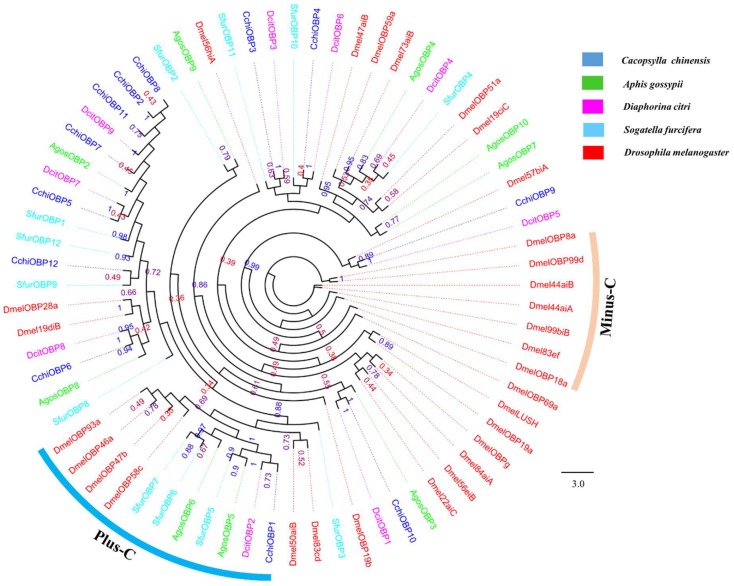
Phylogenetic tree of insect odorant binding proteins (OBPs). The *C. chinensis* translated genes are shown in blue. Abbreviations of other insects are as follows: *Aphis gossypii*, *Diaphorina citri, Sogatella furcifera, Drosophila melanogaster*. This tree was constructed using PhyML based on the alignment results of ClustalX2.0.

**Figure 4 insects-10-00175-f004:**
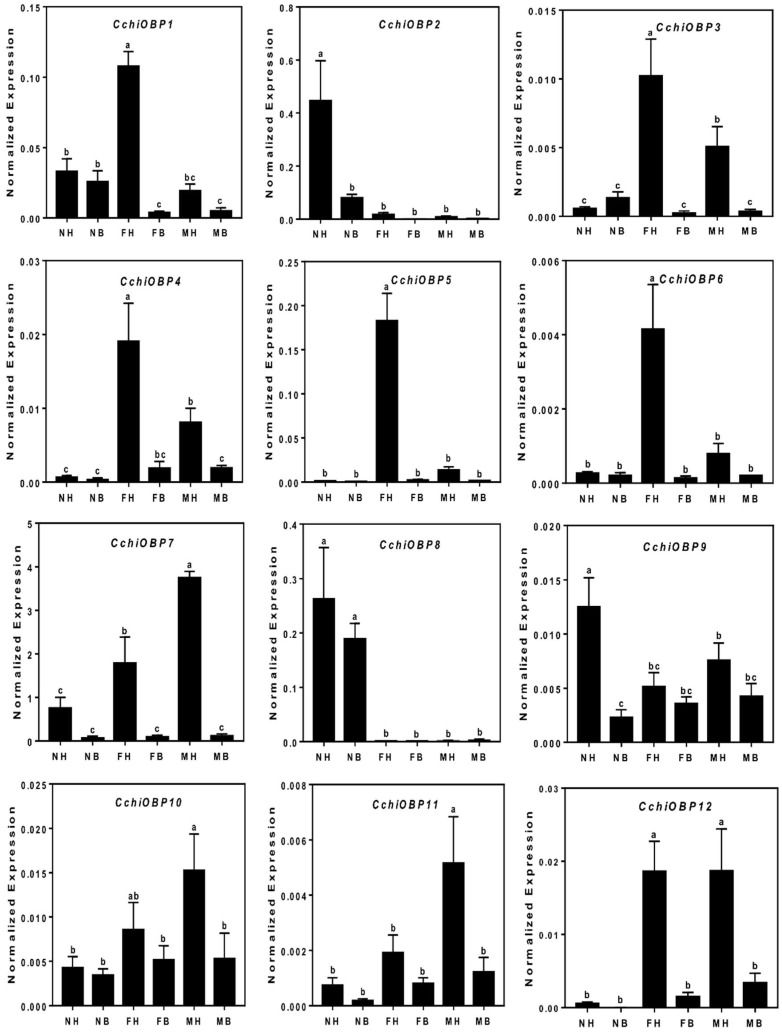
The tissue expression of OBP genes in *C. chinensis*. The different lowercase letters mean significance between tissues (*p* < 0.05, ANOVA, least significance difference (LSD)). NH, nymph head; NB, nymph body; FH, female head; FB, female body; MH, male head; MB, male body.

**Figure 5 insects-10-00175-f005:**
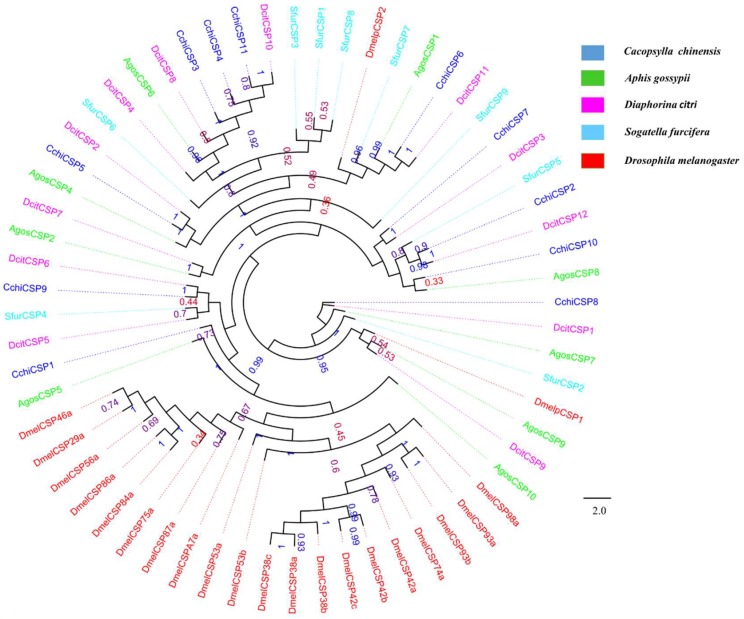
Phylogenetic tree of insect chemosensory proteins (CSPs). The *C. chinensis* translated genes are shown in blue. Other insects are as follows: *Aphis gossypii*, *Diaphorina citri, Sogatella furcifera, Drosophila melanogaster*. This tree was constructed using PhyML based on the alignment results of ClustalX2.0.

**Figure 6 insects-10-00175-f006:**
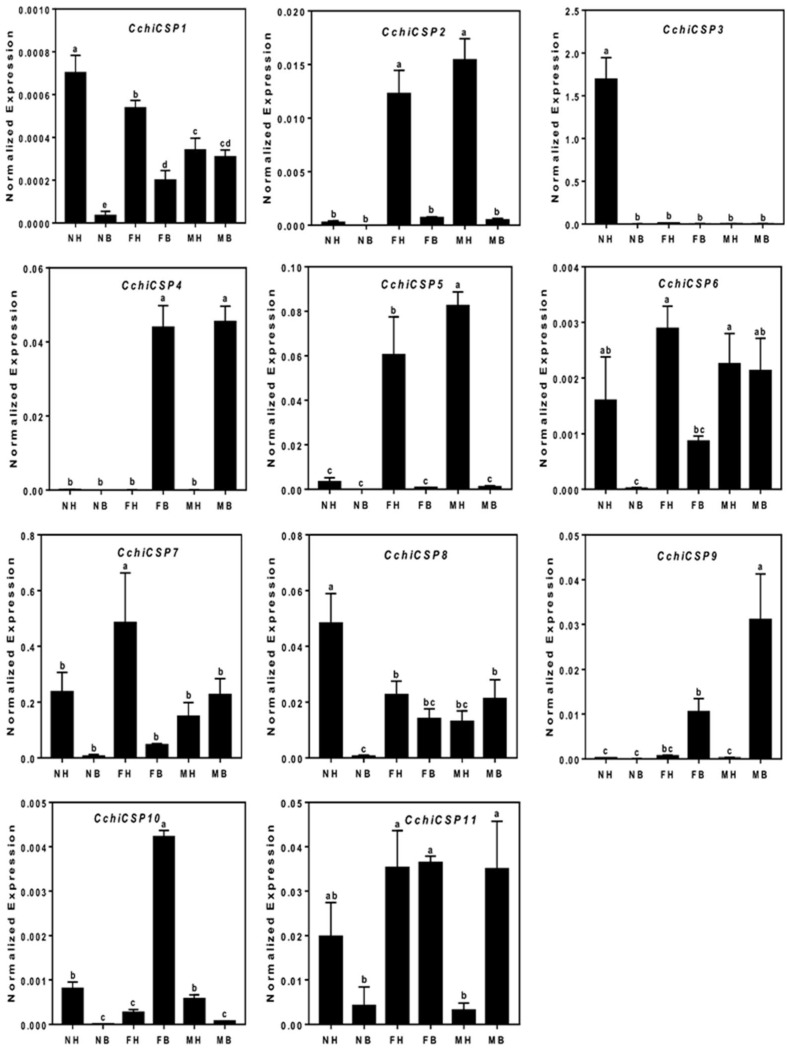
The tissue expression of CSP genes in *C. chinensis*. The different lowercase letters mean significance between tissues (*p* < 0.05, ANOVA, LSD). NH, nymph head; NB, nymph body; FH, female head; FB, female body; MH, male head; MB, male body.

**Figure 7 insects-10-00175-f007:**
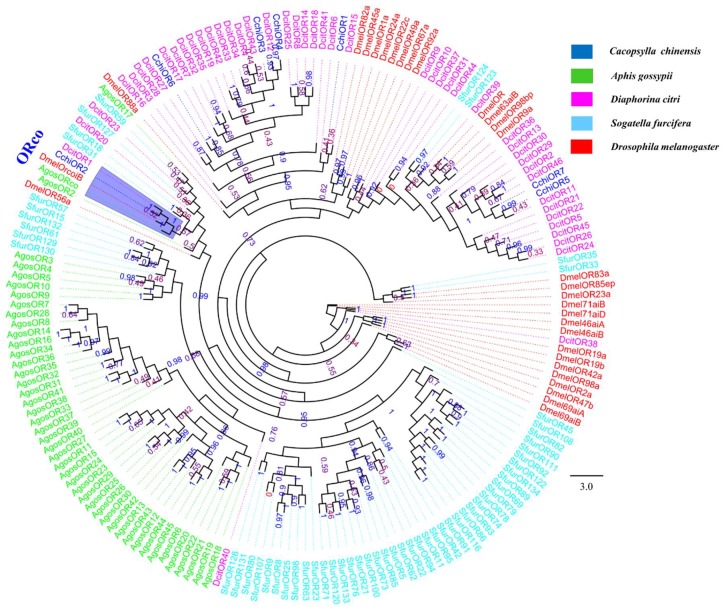
Phylogenetic tree of insect odorant receptors (ORs). The *C. chinensis* translated genes are shown in blue. Other insects are as follows: *Aphis gossypii*, *Diaphorina citri, Sogatella furcifera, Drosophila melanogaster*. Orco clade is marked in blue. This tree was constructed using PhyML based on the alignment results of ClustalX2.0.

**Figure 8 insects-10-00175-f008:**
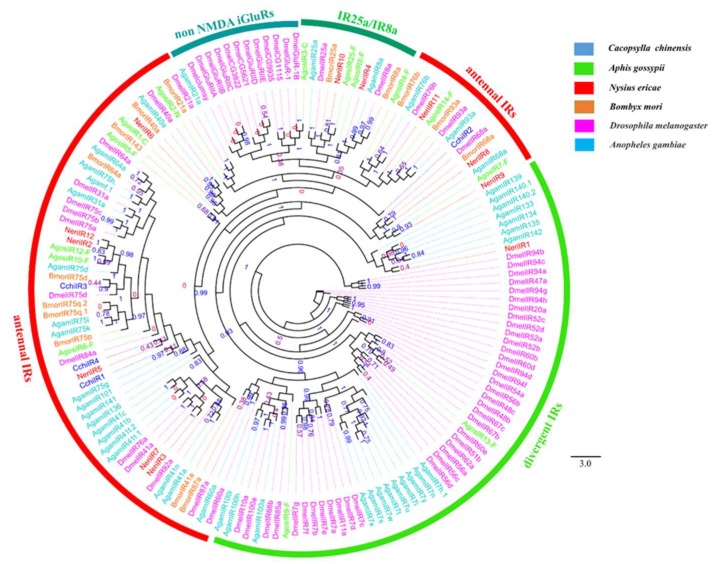
Phylogenetic tree of insect ionotropic receptors (IRs). The *C. chinensis* translated genes are shown in blue. Other insects are as follows: *Aphis gossypii*, *Nysius ericae, Bombyx mori, Drosophila melanogaster, Anopheles gambiae*. This tree was constructed using PhyML based on the alignment results of ClustalX2.0.

**Figure 9 insects-10-00175-f009:**
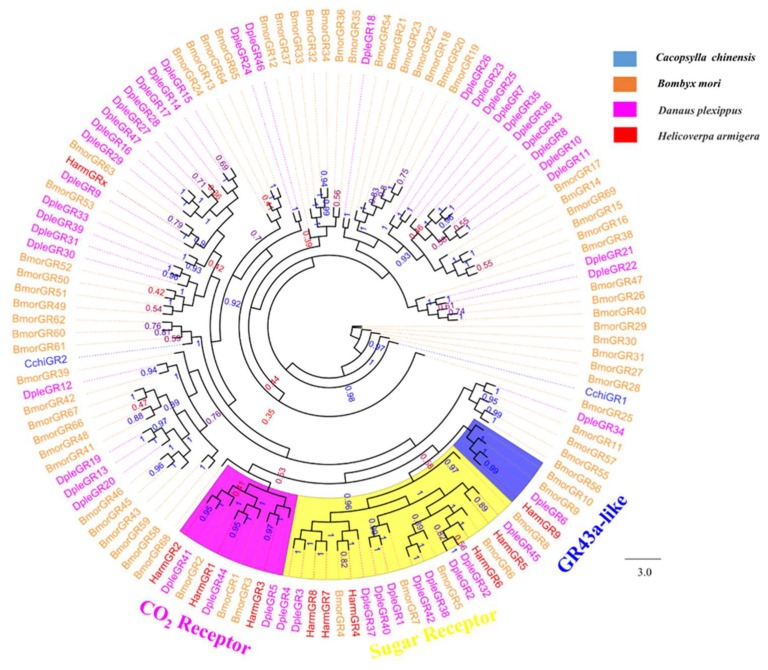
Phylogenetic tree of insect gustatory receptors (GRs). The *C. chinensis* translated genes are shown in blue. Other insects are as follows: *Bombyx mori, Danaus plexippus, Helicoverpa armigera*. This tree was constructed using PhyML based on the alignment results of ClustalX2.0.

**Figure 10 insects-10-00175-f010:**
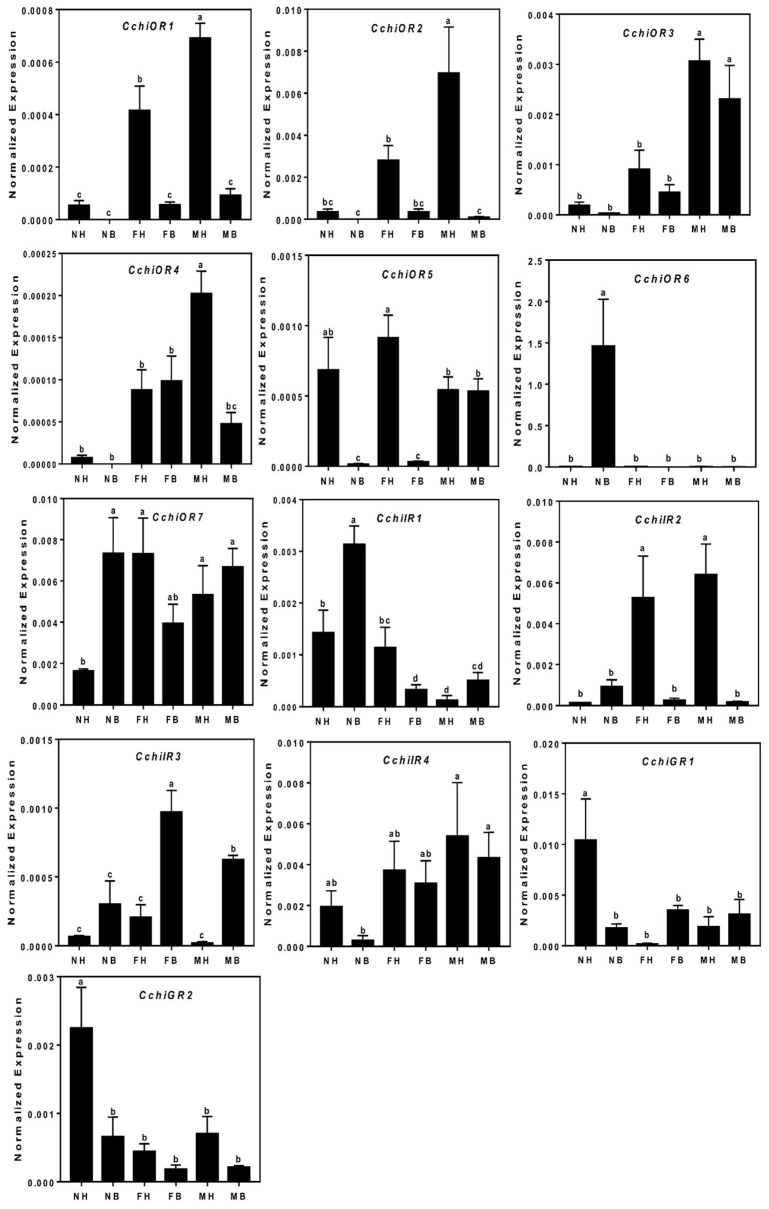
The tissue expression of chemosensory receptor genes in *C. chinensis*. The different lowercase letters mean significance between tissues (*p* < 0.05, ANOVA, LSD). NH, nymph head; NB, nymph body; FH, female head; FB, female body; MH, male head; MB, male body.

**Table 1 insects-10-00175-t001:** Summary of *Cacopsylla chinensis* transcriptome assembly.

Statistics Project	Number
Total clean reads	32,879,148
GC percentage	34.11%
Q20 percentage	99%
Total Unigene nucleotides	40,874,177
Total Unigenes	63,052
N50 of Unigenes (nt)	702
Min length of Unigenes (nt)	201
Mean length of Unigenes (nt)	648.26
Max length of Unigenes (nt)	20,509
Unigenes with homolog in NR	22,392

**Table 2 insects-10-00175-t002:** The Blastx match of *C. chinensis* odorant binding protein (OBP) genes, chemosensory protein (CSP) genes, odorant receptor (OR) genes, ionotropic receptor (IR) genes, and gustatory receptor (GR) genes.

Gene	ORF	Signal	Complete	Best Blastx Match				
Name	(aa)	Peptide	ORF	Name	Acc. No.	Species	E-value	Identity (%)
Odorant Binding Protein (OBP)								
OBP1	238	1–24	Y	odorant-binding protein 5	AHB59658.1	*Sogatella furcifera*	8.00 × 10^−34^	36
OBP2	105	--	N	odorant-binding protein 1	ARR95844.1	*Diaphorina citri*	4.00 × 10^−26^	52
OBP3	65	--	N	general OBP 83a-like	XP_008470659.1	*Diaphorina citri*	1.00 × 10^−226^	65
OBP4	122	1–25	Y	odorant-binding protein 5	ATO59032.1	*Schistocerca gregaria*	8.00 × 10^−13^	37
OBP5	145	1–21	Y	odorant-binding protein 1	ARR95844.1	*Diaphorina citri*	1.00 × 10^−27^	38
OBP6	75	--	N	odorant-binding protein 8	AMD82868.1	*Bemisia tabaci*	8.00 × 10^−18^	56
OBP7	61	--	N	odorant-binding protein 1	ARR95844.1	*Diaphorina citri*	2.00 × 10^−26^	84
OBP8	82	--	N	odorant-binding protein 1	ARR95844.1	*Diaphorina citri*	7.00 × 10^−19^	54
OBP9	135	1–21	Y	general OBP 57c isoform X2	XP_021924930.1	*Zootermopsis nevadensis*	5.00 × 10^−25^	39
OBP10	135	N	Y	odorant-binding protein 3	AGE97633.1	*Aphis gossypii*	2.00 × 10^−13^	32
OBP11	86	--	N	odorant-binding protein 1	ARR95844.1	*Diaphorina citri*	6.00 × 10^−25^	54
OBP12	104	1–24	Y	putative odorant-binding protein A10	XP_008473937.1	*Diaphorina citri*	5.00 × 10^−46^	55
Chemosensory Protein (CSP)								
CSP1	80	--	N	chemosensory protein	AJP61962.1	*Phenacoccus solenopsis*	1.00 × 10^−16^	39
CSP2	104	1–24	Y	chemosensory protein	AVM86436.1	*Corythucha ciliata*	3.00 × 10^−16^	37
CSP3	161	1–16	Y	ejaculatory bulb-specific protein 3-like	XP_008478860.1	*Diaphorina citri*	5.00 × 10^−52^	66
CSP4	103	1–19	Y	ejaculatory bulb-specific protein 3-like	XP_008471453.1	*Diaphorina citri*	2.00 × 10^−60^	75
CSP5	91	--	N	chemosensory protein 1	ARR95843.1	*Diaphorina citri*	1.00 × 10^−38^	74
CSP6	105	--	N	ejaculatory bulb-specific protein 3-like	P_008478140.1	*Diaphorina citri*	1.00 × 10^−64^	89
CSP7	121	1–19	Y	ejaculatory bulb-specific protein 3-like	XP_008473947.1	*Diaphorina citri*	3.00 × 10^−52^	79
CSP8	141	1–21	Y	ejaculatory bulb-specific protein 3-like	XP_018916603.1	*Bemisia tabaci*	1.00 × 10^−38^	79
CSP9	91	--	N	ejaculatory bulb-specific protein 3-like	XP_017300318.1	*Diaphorina citri*	4.00 × 10^−59^	81
CSP10	132	1–19	Y	chemosensory protein	AJP61957.1	*Phenacoccus solenopsis*	1.00 × 10^−22^	41
CSP11	103	1–19	Y	ejaculatory bulb-specific protein 3-like	P_008478860.1	*Diaphorina citri*	2.00 × 10^−74^	89
Odorant receptors (OR)								
OR1	242	--	N	odorant receptor 82a	XP_008477835.1	*Diaphorina citri*	3.00 × 10^−25^	46
OR2	96	--	N	odorant receptor 83b	ADB82908.1	*Loxostege sticticalis*	8.00 × 10^−50^	83
OR3	150	--	N	odorant receptor 85c-like	XP_008476350.1	*Diaphorina citri*	3.00 × 10^−18^	33
OR4	123	--	N	odorant receptor 82a	XP_008477835.1	*Diaphorina citri*	1.00 × 10^−14^	32
OR5	333	--	N	odorant receptor 82a	XP_008477835.1	*Diaphorina citri*	4.00 × 10^−04^	53
OR6	352	--	N	odorant receptor 82a	XP_008477835.1	*Diaphorina citri*	1.00 × 10^−11^	24
OR7	321	--	N	odorant receptor 82a	XP_008477835.1	*Diaphorina citri*	6.00 × 10^−06^	38
Ionotropic receptors (IR)								
IR1	73	--	N	ionotropic receptor 21a isoform X2	XP_024936966.1	*Cephus cinctus*	4.00 × 10^−09^	45
IR2	205	--	N	ionotropic receptor	AUF73076.1	*Anoplophora chinensis*	7.00 × 10^−78^	79
IR3	120	--	N	ionotropic receptor 24	ALD51351.1	*Locusta migratoria*	2.00 × 10^−22^	50
IR4	145	--	N	ionotropic receptor 6	AVH87294.1	*Holotrichia parallela*	1.00 × 10^−36^	59
Gustatory receptor (GR)								
GR1	207	--	N	GR for sugar taste 64f-like	XP_008467720.2	*Diaphorina citri*	2.00 × 10^−84^	78
GR2	200	--	N	gustatory receptor	ABY40623.1	*Tribolium castaneum*	5.00 × 10^−41^	45

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
