# Peer review of "Chemosensory Gene Families in the Oligophagous Pear Pest Cacopsylla chinensis (Hemiptera: Psyllidae)"

_insects, 2019, doi:10.3390/insects10060175_

Round 1

Reviewer 1 Report

Dear editor,

The study by Xu et al entitled “Chemosensory gene families in the oligophagous pear pest Cacopsylla chinensi (Hemiptera: Psyllidae)”(insects-503476-peer-review-v1) analysed the chemosensory gene set of the hemipteran species Cacopsylla chinensis, an oligophagous pest that has become one of the main pests of pear trees in China. To gain first insights how C. chinensis may locate its host at the molecular level, the transcriptome of the head, a major chemosensory organ, was sequenced. 63,052 unigenes were identified with a total of 36 candidate chemosensory genes, including 5 different families: 12 odorant binding proteins (OBPs), 11 chemosensory proteins (CSPs), 7 odorant receptors (ORs), 4 ionotropic receptors (IRs) and 2 gustatory receptors (GRs). Tissue and developmental expression of all genes were further investigated using quantitative real time PCR indicating that some genes displayed male head, female head, or nymph-biased specific/expression. Although the chemosensory genes of C. chinensis identified in this study may provide a valuable resource for future functional analyses of some key genes, to ultimately understand insect host selection of C. chinensis, several unclear and missing aspects render the manuscript not acceptable for publication in its current status. Main issues concern the number of biological replicates used, the unclear transcriptome sequencing setup, the lack of outgroups in the phylogenetic trees and the rather weak interpretation and discussion of the results. Please see my further detailed comments below.

Major issues:

- Material and methods: the number of biological replicates used for qRT-PCR is unclear and needs clarification, see page 2 (“For qRT-PCR analysis, 60 males, 60 females and 60 nymphs were obtained and divided into heads only or bodies only”) versus page 3 (“For six samples, each biological replicate was repeated three times”).

-page 3: “Finally, the cDNA library was sequenced using an Illumina Hiseq 4000 platform”. More sequencing details need to be added, e.g. was paired or single end used? What was the read length etc.?

- “2.4. De novo Assembly and Functional Annotation” is unclear. “A large number of high-quality reads were produced on the sequencing platform, with a low error rate of base calling” sounds imprecise. Where can these results be found? Which version of Trinity was used and what was minimal contig length? The annotation databases (Nr, Swiss-Prot, KEGG, KOG, and COG) are not referenced and it is unclear how they were used, e.g. in any specific order or for certain chemosensory families only?

- 2.5.: How were Signal peptides detected (software, threshold)?

- Figs.3, 5, 6: Outgroups (i.e. insect species from another order than hemipteran) are missing, which is necessary to root the tree. This needs to be done or explained why this is not done. Why were no characterized OBPs such as characterized OBP LUSH from Drosophila melanogaster, crucial for binding pheromones and short-chain alcohols among other compounds, incorporated? (see doi.org/10.1021/bi100540k)

- Discussion: There is too much repetition of mere results, which should be avoided, and too little interpretation and conclusions from the data obtained in this manuscript. To improve discussion, it should begin with the head as main chemosensory organ of this study (e.g. as written by the authors: “As an important olfactory and taste organ, insect heads are important in communication and feeding behavior [45].”) rather than repeating details of results (obtained 63,052 unigenes annotated in the transcriptome of C. chinensis. Among these, 35.51 % or 22,392 had homologs…). Then, other developmental stages should be discussed and compared to other studies. Authors should keep in mind that herbivorous/frugivorous insects rely on distant and contact chemosensory cues and receptors to identify their hosts correctly. Additionally, chemosensory genes are also often found on legs or inner organs (brain, gut) that may govern host plant choice.

Minor issues:

-  General: line numbers are missing and I could not access Supplementary data, unfortunately.

- page 2: “but their [GRs] functional principles are unclear.” Although I agree that there is less known about GRs in comparison to e.g. ORs, there are some recent reviews about insect GR functionality (e.g. Annual Review of Entomology, 2019, 64:227–42; Current Opinion in Neurobiology, 2016, 41:87–91; Natural Product Reports, 2017, 34, 478-483); this should be briefly acknowledged here.

- p. 2.: “and the heads were used for RNA-seq” à How were the heads dissected, e.g. were individuals kept chilled during dissection?

- check all references for correctness (e.g. 2. Lea, must be Leal; update ref 32 since in press from 2016, etc…)

- p.2.: “Three hundred adult and nymph C. chinensis were collected from Dangshan, Suzhou City, China and the heads were used for RNA-seq”. Does this mean that 300 individuals were pooled for one sequencing reaction? Geographical coordinates should also be added here.

- 2.6.: “chemosensory genes in different tissues”. Write the tissues here.

- 2.6.: CchiGAPDH (glyceraldehyde-3-phosphate dehydrogenase) and CchiEF (elongation factor 1-

alpha). Accession numbers (from Genbank) for example need to be added.

- page 4: mentioning of Figure 1 seems out of context

- Figs 3 and 5-8. The bar for bootstrap values is not optimal, since certain values cannot be read clearly enough, e.g. impossible to distinguish 40 from 70 %. Instead, discrete values should be displayed on the branches keeping in mind that only values of at least 75% are reliable indicators for groupings.

- Figure 3: There are subgroups of OBPs, such as classic, minus-C, Plus-C, ABPII (e.g. see Fig.3 doi.org/10.1002/ece3.4246 or doi.org/10.1111/brv.12339), which may give hints on their function. Why were these subclasses not indicated in the tree? At least, they should be briefly mentioned in the discussion.

- Figure 4: font size needs to be increased, since single diagrams are difficult to read. Divide into A, B, C and D to better indicate in which panels one can find expression values for OBPs, CSPs, ORs, IRs. “the different capital letters”. All letters are in lower case, please clarify.

- page 9: “All CchiCSPs shared 37-84 % amino acid identities with other insects”. Do you mean hemipteran species and/or also species from other orders?

- “with at least one moth ortholog distribution on each branch, and higher homology with D. citri”. This is unclear, what moth species do the authors refer to?

Author Response

Respond to Reviewer 1

Comments from the editors and reviewers:
-Reviewer 1

The manuscript summarized the results from transcriptome sequencing in the storage pest, Cacopsylla chinensis. The authors sequence RNA with the Illumina HiSeq and annotate using a variety of online tools, including BLAST2GO and NCBI tools. The main purpose of the paper is to find, annotate, and quantify expression of OBP CSPsORsIRs and GRs.

1. The number of biological replicates used for qRT-PCR is unclear and needs clarification?

Answer: Sorry for our mistake. For qRT-PCR analysis, we separately collected 60 males, 60 females and 60 nymphs divided into heads only or bodies only. They were separately named as MH, MB, FH, FB, NH and NB and placed in nuclease-free centrifuge tubes as one replicate. And each replicate was repeated three times. We have made the corresponding changes in section 2.1 of the revised MS.

2.  Page 3: “Finally, the cDNA library was sequenced using an Illumina Hiseq 4000 platform”. More sequencing details need to be added, e.g. was paired or single end used? What was the read length etc.?

Answer: Thank you for this comment. The cDNA library was sequenced using an Illumina Hiseq 4000 platform with the paired-end (PE). The sequencing read length was PE150bp. After sequencing, we gained 32,879,148 clean reads. We made the corresponding changes in section 2.3 of the revised MS. About a description of the reads, please see the section 3.1 and Table 1.

3. De novo Assembly and Functional Annotation” is unclear. “A large number of high-quality reads were produced on the sequencing platform, with a low error rate of base calling” sounds imprecise. Where can these results be found? Which version of Trinity was used and what was minimal contig length? The annotation databases (Nr, Swiss-Prot, KEGG, KOG, and COG) are not referenced and it is unclear how they were used, e.g. in any specific order or for certain chemosensory families only?

Response: (1) Thank you for your suggestions. We have changed the “De novo Assembly and Functional Annotation” to “De novo Assembly and Unigene Annotation”, and we revised “A large number of high-quality reads were produced on the sequencing platform, with a low error rate of base calling” as follows: After using sequencing by synthesis (SBS) technology, raw reads were cleaned by removing adaptor reads, ambiguous reads (‘N’ > 10%), and low-quality reads (that is, where more than 50% of bases in a read had a quality value Q 5) using perl script. Clean reads were then de novo assembled using the Trinity program (v2.4.0) with default parameters [42].

(2) The reads of C. chinensis have been deposited (accession number: SRA9127897).

(3) The version of Trinity is v2.4.0, and the minimal contig length is 301bp.

(4) The annotation databases (Nr [43], Swiss-Prot [44], KEGG [45], KOG [46], and COG [47]) were used to obtain amino acid sequences of Unigene genes. And then, HMMER [48] software was used to compare with Pfam [49] database (including chemosensory proteins families in different insects) to obtain annotation information of Unigene.

We also have made the corresponding changes in the section 2.4, please see the revised MS.

References:

42. Haas BJ, Papanicolaou A, Yassour M, Grabherr MG, Blood PD, Bowden J, Couger MB, Eccles D, Li B, Lieber M: De novo transcript sequence reconstruction from RNA-seq using the Trinity platform for reference generation and analysis. Nature Protocols 2013, 8:1494-1512.

43. Deng YY, Li JQ, Wu SF:  Integrated  nr  Database  in  Protein  Annotation  System  and  Its Localization. Computer Engineering 2006, 32:71 -74.

44. Apweiler R, Bairoch A, Wu CH, Barker WC, Boeckmann B, Ferro S, Gasteiger E, Huang H, Lopez R, Magrane M: UniProt: the universal protein knowledgebase. Nucleic acids Res 2004, 32:D115-D119.

45. Kanehisa M, Goto S, Kawashima S, Okuno Y, Hattori M: The KEGG resource for deciphering the genome. Nucleic acids Res 2004, 32:D277-D280.

46. Jensen LJ, Julien P, Kuhn M, Mering C, Muller J, Doerks T, Bork P: eggNOG: automated construction and annotation of orthologous groups of genes. Nucleic acids Res 2008, 36:D250-D254.

47. Tatusov RL, Galperin MY, Natale DA, Koonin EV: The COG database: a tool for genome-scale analysis of protein functions and evolution. Nucleic acids Res 2000, 28:33-36.

48. Zhang Z, Wood W: A profile hidden Markov model for signal peptides generated by HMMER. Bioinformatics 2003, 19:307-308.

49. Bateman A, Coin L, Durbin R, Finn R, Hollich V, Griffiths‐Jones S, Khanna A, Marshall M, Moxon S, Sonnhammer E: The Pfam protein families database. Nucleic acids research 2004, 32:D138-D141.

4. How were Signal peptides detected (software, threshold)?

Answer: Sorry for missing this information. We used SignalP 4.1 server (Nielsen H, Methods Mol Biol, 2017) to predict the Signal peptides of OBPs and CSPs with default parameters, and the information have been added in section 2.5 of the revised MS. The threshold includes Range, Weight, and D cut. The ‘Range’ is the number of amino acid positions before and after a potential cleavage site from which the mean signal peptide likeliness is calculated, and the value for Eukaryotes is 20-24. ‘Weight’ is the term used in the linear combination of Smean and Ymax, and the value for Eukaryotes is 0.41-0.54. ‘D cut’ is the optimized threshold for the prediction of a potential signal peptide, and the value for Eukaryotes is 0.36-0.51.

Reference:

Nielsen H: Predicting Secretory Proteins with SignalP. Methods Mol Biol 2017, 1611:59-73.

5. Figs.3, 5, 6: Outgroups (i.e. insect species from another order than hemipteran) are missing, which is necessary to root the tree. This needs to be done or explained why this is not done. Why were no characterized OBPs such as characterized OBP LUSH from Drosophila melanogaster, crucial for binding pheromones and short-chain alcohols among other compounds, incorporated? (see doi.org/10.1021/bi100540k)

Answer: Sorry for our mistake. As your suggestion, we have reconstructed the phylogenetic trees (the order of new Figs is: 3, 5, and 7) with Drosophila melanogaster as outgroup. The new OBP tree includes the LUSH gene of D. melanogaster, but no OBP of C. chinensis is clustered with LUSH. Please see the Figs. 3, 5, and 7 of revised MS.

6. Discussion: There is too much repetition of mere results, which should be avoided, and too little interpretation and conclusions from the data obtained in this manuscript. To improve discussion, it should begin with the head as main chemosensory organ of this study (e.g. as written by the authors: “As an important olfactory and taste organ, insect heads are important in communication and feeding behavior [45].”) rather than repeating details of results (obtained 63,052 unigenes annotated in the transcriptome of C. chinensis. Among these, 35.51 % or 22,392 had homologs…). Then, other developmental stages should be discussed and compared to other studies. Authors should keep in mind that herbivorous/frugivorous insects rely on distant and contact chemosensory cues and receptors to identify their hosts correctly. Additionally, chemosensory genes are also often found on legs or inner organs (brain, gut) that may govern host plant choice.

Answer: Thank you for this suggestion. We have reorganized the section and added some information to the revised Discussion.

Some specifics:

General: line numbers are missing and I could not access Supplementary data, unfortunately.

Answer: Sorry for our mistake, and we have added the line number in the revised MS. However, we have submitted supplementary materials together with the MS through an online submission system of the journal. We will resubmit the supplementary materials.

page 2: “but their [GRs] functional principles are unclear.” Although I agree that there is less known about GRs in comparison to e.g. ORs, there are some recent reviews about insect GR functionality (e.g. Annual Review of Entomology, 2019, 64:227–42; Current Opinion in Neurobiology, 2016, 41:87–91; Natural Product Reports, 2017, 34, 478-483); this should be briefly acknowledged here.

Answer: As your suggestion, we have added the recent reviews about insect GR functionality in the revised MS.

p. 2: “and the heads were used for RNA-seq” à How were the heads dissected, e.g. were individuals kept chilled during dissection? check all references for correctness (e.g. 2. Lea, must be Leal; update ref 32 since in press from 2016, etc…)

Answer: (1) The head dissected out under stereomicroscope with sterile scalpel, were immediately mixed into liquid nitrogen and sent to Genepioneer Biotech Corporation (Nanjing, China) for RNA-seq. We have made the corresponding changes in section 2.1 of the revised MS.

(2) Sorry for our mistake, and we have checked and corrected all reference errors.

p.2.: “Three hundred adult and nymph C. chinensis were collected from Dangshan, Suzhou City, China and the heads were used for RNA-seq”. Does this mean that 300 individuals were pooled for one sequencing reaction? Geographical coordinates should also be added here.

Answer: Three hundred (100 males, 100 females and 100 nymphs) of C. chinensis were collected from Dangshan (N 34° 27ʹ 42.62ʺ, E116° 31ʹ 40.72ʺ), Suzhou City, China and the heads were used for RNA-seq”. Yes, 300 individuals were pooled for one sequencing reaction. We have made the corresponding changes in section 2.1 of the revised MS.

2.6.: “chemosensory genes in different tissues”. Write the tissues here.

Answer: Thank you for this suggestion. We have added all the tissue names in the revised MS. The sentence “chemosensory genes in different tissues” have been changed to “chemosensory genes in different tissues (MH, MB, FH, FB, NH and NB)”

2.6.: CchiGAPDH (glyceraldehyde-3-phosphate dehydrogenase) and CchiEF (elongation factor 1- alpha). Accession numbers (from Genbank) for example need to be added.

Answer: As your suggestion, we have added the accession numbers (from Genbank) (CchiGAPDH: MK940861, CchiEF: MK940862) in the revised MS.

page 4: mentioning of Figure 1 seems out of context

Answer: Because there are few studies on C. chinensis at present, in order to make readers better know this insect, we showed its morphological picture in Figure 1. As your suggestion, we have moved Figure 1 from section 3.1 to 2.1 in the revised MS.

Figs 3 and 5-8. The bar for bootstrap values is not optimal, since certain values cannot be read clearly enough, e.g. impossible to distinguish 40 from 70 %. Instead, discrete values should be displayed on the branches keeping in mind that only values of at least 75% are reliable indicators for groupings.

Answer: As your suggestion, we have amended these Figures in the revised MS.

Figure 3: There are subgroups of OBPs, such as classic, minus-C, Plus-C, ABPII (e.g. see Fig.3 doi.org/10.1002/ece3.4246 or doi.org/10.1111/brv.12339), which may give hints on their function. Why were these subclasses not indicated in the tree? At least, they should be briefly mentioned in the discussion

Answer: Thank you for this suggestion. We have added the information of OBP subgroups in the revised Figure 3. The results found only CchiOBP1 belong to the Plus-C subgroup, others CchiOBPs were clustered in the Classic subgroup.

- Figure 4: font size needs to be increased, since single diagrams are difficult to read. Divide into A, B, C and D to better indicate in which panels one can find expression values for OBPs, CSPs, ORs, IRs. “the different capital letters”. All letters are in lower case, please clarify.

Answer: (1) Thank you for this suggestion. We have modified the Figure 4 and divide it into Figure 4, 6 and 10 in the revised manuscript. (2) Sorry for our mistake, “the different capital letters” have been changed to “the different lower case letters” in the revised MS.

- page 9: “All CchiCSPs shared 37-84 % amino acid identities with other insects”. Do you mean hemipteran species and/or also species from other orders?

Answer: According to the Blast results (Table 2), all CchiCSPs shared 37-89% amino acid identities with other Hemipteran insects. We have made the corresponding changes in section 3.4 of the revised MS.

“with at least one moth ortholog distribution on each branch, and higher homology with D. citri”. This is unclear, what moth species do the authors refer to?

Answer: Sorry for our mistake, the sentence have been changed to “with one D. citri ortholog distribution on each branch”. Please see the revised MS.

Reviewer 2 Report

Comments and Suggestions for Authors

This manuscript provides information on chemosensory gene families and their potential functions in pear pest, Cacopsylla chinensi using Illumina sequencing

The paper is well written except for minor changes/corrections.

In several places of the manuscript, CchiCSPs, CchiIRs, GRs, ORs, OBPs are not italicized. Please double-check.

The references are not double-checked. There are several typos where most of the scientific names are not italicized (References- 7,18, 20, 25, 35, 49, 50, 51, 53, 58, etc.).

-       Reference 3 and 23 are exactly the same.

-       Check the alignment of reference 46 and font of 48 where ‘r’ in glover is bold.

-       Be consistent with journal names (Eg. PloS One)

Figure 1 is not necessary and does not explain anything related to the results in the manuscript. The first line of results where Figure 1 is mentioned is not relevant to the context, so this can be removed from the manuscript.

Please refer to the line specific comments below

·      Please rephrase the last sentence of the introduction

·      Check font size of ‘2.1.’ and in this section briefly explain how the heads were dissected. In buffer? Using forceps? More information is needed.

·      The samples were preserved in liquid nitrogen until RNA extraction? Mention.

·      Please rephrase all the sentences that start with ‘qRT-PCR’

·      Section 2.4 – After the raw reads ‘were’ filtered

·      Bold ‘Table 1’ – be consistent with other figure legends

·      Table 1 – Replace ‘Total unigene’ with ‘Total unigenes’

·      Section 3.3 – Include space between Chemosensory Gene

·      Figure 3 – remove ‘Agoss’

·      Bold ‘Table 2’ and replace ‘Match’ with ‘match’ in the title

·      Be consistent - mention Horvath with Spodoptera furcifera

·      Figure 6 – ORco or Orco?

·      Last line of discussion – in vivo and in vitro methods such as? Briefly mention.

Author Response

Respond to Reviewer 2

Reviewer 2

In several places of the manuscript, CchiCSPs, CchiIRs, GRs, ORs, OBPs are not italicized. Please double-check.

Answer: Thank you for pointing out these errors. We have checked and corrected all the errors throughout the MS. Abbreviations for nucleic acid in italics and for amino acids or proteins in non-italics.

The references are not double-checked. There are several typos where most of the scientific names are not italicized (References- 7, 18, 20, 25, 35, 49, 50, 51, 53, 58, etc.).

Answer: We apologize for these errors. We have checked and corrected all the errors throughout the MS.

- Reference 3 and 23 are exactly the same.

Answer: Sorry for our mistake, we have removed the reference 3 in the revised MS.

- Check the alignment of reference 46 and font of 48 where ‘r’ in glover is bold.

Answer: We apologize for this matter. The “glover” have been changed to “Glover” in the revised MS.

- Be consistent with journal names (Eg. PloS One)

Answer: “PLoS One” is right, and we have checked and corrected all the incorrect reference format.

Figure 1 is not necessary and does not explain anything related to the results in the manuscript. The first line of results where Figure 1 is mentioned is not relevant to the context, so this can be removed from the manuscript.

Answer: Because there are few studies on C. chinensis at present, in order to make readers better know this insect, we showed its morphological picture in Figure 1. We moved Figure 1 from section 3.1 to 2.1 in the revised MS to give readers a clear understanding of which tissues were used in this study.

Please refer to the line specific comments below

Please rephrase the last sentence of the introduction

Answer: The sentence have been changed to “We first conducted heads transcriptome analysis of C. chinensis. Then further analyzed the phylogenetic trees and examined tissue expression of the chemosensory genes”.

Check font size of ‘2.1.’ and in this section briefly explain how the heads were dissected. In buffer? Using forceps? More information is needed.

Answer: We have checked and corrected all errors in different font formats. The head dissected out under stereomicroscope with sterile scalpel, were immediately mixed into liquid nitrogen and sent to Genepioneer Biotech Corporation (Nanjing, China) for RNA-seq. We have made the corresponding changes in section 2.1 of the revised MS.

The samples were preserved in liquid nitrogen until RNA extraction? Mention.

Answer: Yes, we have added the description in section 2.1 of the revised MS.

Please rephrase all the sentences that start with ‘qRT-PCR’

Answer: Thank you for your suggestion, and we have checked and corrected all the sentences that start with ‘qRT-PCR’. Please see the revised MS.

Section 2.4 – After the raw reads ‘were’ filtered

Bold ‘Table 1’ – be consistent with other figure legends

Table 1 – Replace ‘Total unigene’ with ‘Total unigenes’

Section 3.3 – Include space between Chemosensory Gene

Figure 3 – remove ‘Agoss’

Answer: Thank you for pointing out these mistakes. We have checked and corrected all the errors.

Round 2

Reviewer 1 Report

Dear Editor,

the authors have done a good job in revising the manuscript. I have only a few comments left:

- The resolution of Figs. 3, 5, 7, 8 and 9 should be increased, since it is currently too low.

- Line 253: delete “,. “ before “Therefore”. Also, “an important tissue of“, should be substituted with „important in“.

- Table S1: Caption: write insect species and list species in caption since the table is very long.

- Please check English grammar and spelling throughout manuscript.

Author Response

- The resolution of Figs. 3, 5, 7, 8 and 9 should be increased, since it is currently too 

low.   

Answer: Thanks for your suggestion, and we have done our best to improve the pixels 

of these pictures. Please see the revised MS. 

- Line 253: delete “,. “ before “Therefore”. Also, “an important tissue of“, should be   

substituted with „important in“.   

- Table S1: Caption: write insect species and list species in caption since the table is 

very long.   

- Please check English grammar and spelling throughout manuscript.   

Answer:  Thanks  for  your  suggestion.  We  have  corrected  the  two  errors  and  also 

checked the English grammar and spelling throughout manuscript.